# Usefulness of Contrast-Enhanced Endoscopic Ultrasound (CH-EUS) to Guide the Treatment Choice in Superficial Rectal Lesions: A Case Series

**DOI:** 10.3390/diagnostics13132267

**Published:** 2023-07-04

**Authors:** Giulia Gibiino, Monica Sbrancia, Cecilia Binda, Chiara Coluccio, Stefano Fabbri, Paolo Giuffrida, Graziana Gallo, Luca Saragoni, Roberta Maselli, Alessandro Repici, Carlo Fabbri

**Affiliations:** 1Gastroenterology and Digestive Endoscopy Unit, 47121 Forlì, Italy; giulia.gibiino@gmail.com (G.G.); monica.sbrancia@auslromagna.it (M.S.); colucciochiara@gmail.com (C.C.); stefano.fabbri20@gmail.com (S.F.); paologiuffrida1@gmail.com (P.G.); carlo.fabbri@auslromagna.it (C.F.); 2Section of Gastroenterology & Hepatology, Department of Health Promotion Sciences Maternal and Infant Care, Internal Medicine and Medical Specialties, PROMISE, University of Palermo, 90127 Palermo, Italy; 3Pathology Department, M. Bufalini Hospital, Emilia Romagna, 47521 Cesena, Italy; graziana.gallo@auslromagna.it; 4Pathology Department, G.B. Morgagni-L. Pierantoni Hospital, Emilia-Romagna, 47121 Forlì, Italy; luca.saragoni@auslromagna.it; 5Endoscopy Unit, Humanitas Clinical and Research Center-IRCCS, 20089 Rozzano, Italy; roberta.maselli@hunimed.eu (R.M.); alessandro.repici@hunimed.eu (A.R.)

**Keywords:** rectum polyps, endoscopic submucosal dissection, rectum staging, tumor angiogenesis

## Abstract

Introduction: Large rectal lesions can conceal submucosal invasion and cancer nodules. Despite the increasing diffusion of high-definition endoscopes and the importance of an accurate morphological evaluation, a complete assessment in this setting can be challenging. Endoscopic ultrasound (EUS) plays an established role in the locoregional staging of rectal cancer, although this technique has a tendency toward the over-estimation of the loco-regional (T) staging. However, there are still few data on contrast-enhanced endoscopic ultrasound (CH-EUS), especially if this ancillary technique may increase the accuracy for predicting invasive nodules among large rectal lesions. Material and Methods: Consecutive large (≥20 mm) superficial rectal lesions with high-definition endoscopy, characterized by focal areas suggestive for invasive cancer/2B type according to JNET classification, were considered for additional standardized evaluation via CH-EUS with Sonovue ©. Results: From 2020 to 2023, we evaluated 12 consecutive superficial rectal lesions with sizes ranging from 20 to 180 mm. This evaluation provided additional elements to support the therapeutic decision made. Lesions were treated with surgical (3/12) or endoscopic treatment (9/12) according to their morphology and CH-EUS evaluation. Conclusion: Contrast-enhanced endoscopic ultrasound can provide an additional evaluation for large and difficult-to-classify rectal lesions. In our experience, CH-EUS staging corresponded to the final pathological stages in 9/12 (75%) lesions, improving the distinction between T1 and T2 lesions. Larger prospective studies and randomized trials should be conducted to support and standardize this approach.

## 1. Introduction

Colorectal cancer causes 700,000 deaths every year, being the fourth cause of cancer death worldwide; therefore, endoscopy plays a key role in the early diagnosis and removal of these lesions [1]. In the rectum, endoscopic resection can be performed via Endoscopic Mucosal Resection (EMR) or Submucosal Dissection (ESD) in addition to the viable alternative of minimally invasive transanal surgical techniques [2,3,4,5]. Endoscopic resection with ESD is curative in the case of neoplasia with a low risk of nodal metastasis with shallow T1 (SM) invasion ((<1000 micron, SM1), without unfavorable factors (low differentiation—G3, lympho-vascular invasion and tumor budding).

According to the decision algorithm proposed by the current European Guidelines [6,7,8,9,10,11], en-bloc EMR, ESD or surgery are indicated in rectal lesions after an accurate evaluation in terms of size, morphology, location, margin evaluation and vascular and surface pattern. For all the cases located in the rectum showing pattern NICE 3 ore JNET 3 [12,13,14] with the suspicion of deep submucosal invasion, endoscopic ultrasound (EUS) or Magnetic Resonance Imaging (MRI) should be performed. The assessment of the susceptibility to endoscopic resection of neoplastic lesions of the colo-rectum should not be based on histological data from biopsies which may cause fibrosis or interfere with the lifting of the submucosa during endoscopic resection and do not always reflect the degree of the major carcinomatous component depending on the exact location of sampling [15].

Endoscopic ultrasound (EUS) plays a role in the locoregional staging of rectal cancer both for the evaluation of the degree of infiltration (T) and mostly for lymph node assessment staging (N) [8]. The accuracy of the EUS in T staging is, however, extremely variable, ranging from 63% to 96% depending on the study [16,17,18,19,20]. A review published by Puli et al. in 2009 [21] showed that EUS had sensibility and specificity, respectively, of 87.8% and 98.3% for T1 rectal lesions. However, a subsequent prospective German study on 7096 patients in 300 centers showed that the correspondence between echoendoscope staging (uT) and anatomopathological staging (pT) was 64.7%. However, in 18% of cases, EUS underestimated the degree of infiltration, while in 17.3%, it overestimated locoregional staging. Stages T2 and T4 were associated with lower accuracy than T1 and T3 lesions. According to the same authors, accuracy was also directly related to the degree of experience and to the center volume. Even though EUS and MRI both play a role in rectal tumor staging with a similar accuracy to T and N evaluation [22], their position in the evaluation of superficial and early lesions is still not defined. Although studies with miniprobes revealed that EUS can be useful in defining proper staging of rectal lesions, helping to define between patients eligible for endoscopic resections and surgical patients [23], the accuracy in distinguishing between T1a and T1b lesions is still unsatisfactory.

However, the advent of ultrasound image enhancement systems, such as ultrasound contrast agents, brought a valuable aid for improving the definition of the degree of parietal infiltration. Contrast-enhanced harmonic ultrasound (CH-EUS) uses a micro-bubble-based solution which, once injected into the bloodstream, breaks or refracts the ultrasound signal coming from the probe, generating an amplified acoustic signal. Today, the most widely used ultrasound contrast agent, both in trans-abdominal ultrasound and in EUS, is SonoVue © (sulfur hexafluoride MBs; Bracco Inter-national BV, Amsterdam, The Netherlands), which is composed of sulfur particles coated with a phospholipid membrane.

While the role of CH-EUS has been widely studied at the level of pancreatic lesions and submucosal lesions, both in terms of detection and differential diagnosis [24], its role in the evaluation of mucosal lesions of the rectum is a field almost completely unexplored. To date, there is only one study that has used CH-EUS with a rigid probe for the evaluation of rectal tumors to define the degree of neoangiogenesis, the biological aggression and clinical outcome [25]. We report our experience showing that CH-EUS can increase the staging accuracy in suspected T1 tumors of the rectum to guide the management of these patients and ensure a curative resection.

## 2. Materials and Methods

### 2.1. Patients’ Selection

Consecutive patients with rectal lesions ≥20 mm and with the following characteristics were included:
-Paris and Laterally Spreading Tumor (LST) classification corresponding to 0-Is, 0-IIa, 0-IIb or mixed lesions (0-IIc + IIa or 0-IIa + IIc); LST nongranular (LST-NG or pseudodepressed, IIc); or LST granular nodular mixed type.-The concomitant presence of a demarcated focal area depressed or with an irregular surface pattern or bulky component corresponding to JNET 2B/3 (Figure 1).

### 2.2. Contrast-Enhanced Endoscopic Ultrasound Technique

All suspected lesions were evaluated during the same session with a linear endoscopic ultrasound [26]. Patients received an informed consent form including details about the exam and contrast agent administration.

The evaluation was performed by two experienced endoscopists in EUS. The endoscopic evaluation was performed under conscious sedation using the same high-resolution processor. Following the B-mode examination of the rectal lesions, CH-EUS was performed using Sonovue © 2.4 mL. The contrast medium was activated with 45 s of agitation and then injected into a cubital vein. After the injection, 10 mL of saline was administered. The ultrasound machine was set with an MI of 0.21. After the contrast administration, arterial, venous and late uptake were assessed at 30 s, 60 s and 90 s, respectively.

Specifically, EUS examination was standardized as follows:
-Standard EUS for the definition of the degree of parietal infiltration without contrast agent and the search for pathological lymph nodes providing staging.-The injection of the intravenous contrast agent (Sonovue ©) and the assessment of the entirety of the various layers of the rectum wall arterial and portal venous phases (Figure 2 and Figure 3). Finally, a uT and N stage was assigned for each lesion. After the removal of the lesions via an endoscopic or surgical procedure, complete pathological staging (pTNM) was performed, including the degree of differentiation, lymphatic invasion, vascular invasion and tumor budding according to the WHO classification [27].

### 2.3. Statistical Analysis

Descriptive statistics for nonparametric distribution were used. Means, medians and standard deviations were used to report the results, as appropriate.

## 3. Results

From January 2020 to January 2023, a total of 12 patients were included (M/F 5/7, mean age 70.4 ± 6.5 years). Indications for CH-EUS comprised rectal lesions with a median size of 35 mm (20–180). The baseline characteristics are reported in Table 1. We did not observe any early or delayed adverse events following the administration of intravenous contrast agent. In three cases, the CH-EUS showed an invasive pattern, and the patients were submitted to a surgical approach, mainly represented by anterior resection. In eight cases, CH-EUS confirmed an early stage (uTis or uT1) and curative endoscopic resection was performed (Figure 4). In 9 out of 12 patients (75%), the definitive pathological diagnosis corresponded to the initial CH-EUS staging. A single case of overstaging was observed for a voluminous lesion of the rectum which, due to severe electrolyte alterations, was identified as Mckittrick–Wheelok Syndrome [28] (Figure 5). Based on morphology and MRI and CH-EUS staging, an indication was placed for ESD, which was later converted to surgery due to the protracted procedure and the patient’s delicate condition. In two other cases of adenoma of the distal rectum, evaluation with CH-EUS resulted in overstaging.

## 4. Discussion

We present the first case series of rectal lesions where the use of contrast agent during endoscopic ultrasound is proposed as an additional tool to support therapeutic decision making. Large rectal lesions can be difficult to examine properly even in expert hands [16]. Lesions’ morphologies should guide therapeutic decision, but it was shown that the agreement is only moderate, even among experts [29]. Furthermore, training in the use of the various advanced imaging methods is not standardized, particularly in Western countries [30].

Evaluation with EUS and microvascular pattern distribution with contrast enhancement can be performed in the same endoscopy session, without any additional bowel preparation. It can be targeted on individual areas that are more suspicious and worthy of study without having to perform a global image, as is the case with MRI. Rectal cancer upon EUS evaluation appears as a hypoechoic mass where the regular stratification wall is altered. T1 lesions of the rectum are limited to the mucosa and submucosa (up to the third layer), with no signs of extension to the fourth layer. T2 lesions infiltrate the muscle layer (IV stratum) but do not pass through the serosa (V stratum). A first attempt of the classification of submucosal infiltration was proposed by Hurlstone et al. in 2007 using a high-frequency miniprobe and compared to histopathology [23]: SM1: hypoechoic mass localized within the first hypoechoic layer; SM2: hypoechoic mass infiltrating the second hypoechoic layer; SM3: hypoechoic mass infiltrating the third layer ± extension to the hyperechoic muscularis. However, this study has been largely criticized and has not been confirmed by further studies. Various attempts have subsequently been made considering the additional diagnostic yield provided by a contrast agent [31,32,33]; however, to date, there are no recommendations or standardized approaches in this setting. In our study population, the use of a contrast agent allowed an improvement in therapeutic decisions according to the predicted submucosal invasion with good correspondence with the final staging. A single case of understaging was observed in a patient affected by Mckittrick–Wheelok Syndrome. In this specific setting, the considerable size of the lesion could constitute the very limitation for the application of EUS. Two cases of adenoma were overstaged as T1 carcinomas, but it is noteworthy that both lesions could have been altered by fibrosis: in one case by pre-resection biopsy, and in the other case, the lesion was a recurrence of previous piecemeal excision.

Beyond adenomas with dysplasia, the addition of CH-EUS proved beneficial overall in distinguishing T1 and T2 lesions. The advancement of certain endoscopic techniques could allow the use of CH-EUS even more focused on a technical study of the lesion to choose which endoscopic technique is most suitable. The first Endoscopic Intermuscular Dissection (EID) feasibility and safety study on cases of rectal lesions with suspected deep submucosal invasion was recently published [34]. This advance has shed light on the persistent problem of the absence of optical or MRI or EUS classifications able to distinguish T1Sm2/3 from T2/T3 cancers with certainty [35]. The proposal to introduce CH-EUS in the study of lesions of the rectum could create new perspectives in this regard.

We are aware that this study has many limitations. Firstly, it is a retrospective case series, and individual cases were submitted to CH-EUS not systematically but only in the case of doubt due to the size and heterogeneity of the lesion. Furthermore, all lesions were studied with a linear echoendoscope, as this is proven to be non-inferior to a radial one [26]. Clearly, the accuracy of CH-EUS in the diagnosis of these lesions in terms of T-stage is yet to be defined, as well as the degree of training and experience required. We believe that our contribution can be the beginning of further studies based on the expert use of the CH-EUS, aiming toward a better selection of T1 and T2 rectal adenocarcinomas without having to introduce additional risks.

## 5. Conclusions

In our experience, performing CH-EUS in the evaluation of voluminous and complex rectal lesions in experienced hands could improve accuracy in rectal cancer staging, giving a guide for the therapeutic decision, without any additional risk for the patients. CH-EUS staging can provide useful information regarding either the integrity of the muscular layer and the presence of vascularization, which are both factors known to be predictive for non-curative endoscopic resection. The limited experience in this first case series does not allow for a definition of accuracy, but it shows encouraging results, as it allowed a clinically successful therapeutic decision in the majority of cases.

The dissemination of this method for these lesions could lead to a future definition of the agreement between experts and thus a shared classification to define the degree of deep infiltration. Pre-interventional CH-EUS could promote the adoption of intermuscular dissection techniques or more advanced resections beyond the submucosal plane, avoiding surgical overtreatment. Large, prospective studies are needed to define the accuracy of this method in this setting, also with respect to the evolution of endoscopic therapy techniques.

## Figures and Tables

**Figure 1 diagnostics-13-02267-f001:**
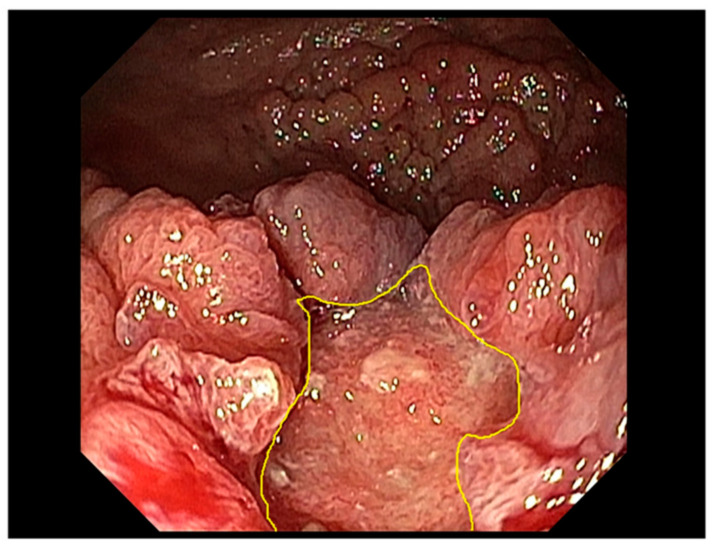
A large LST-GM of the distal rectum, showing a focal area with suspicious invasive pattern (type 3 according to JNET classification).

**Figure 2 diagnostics-13-02267-f002:**
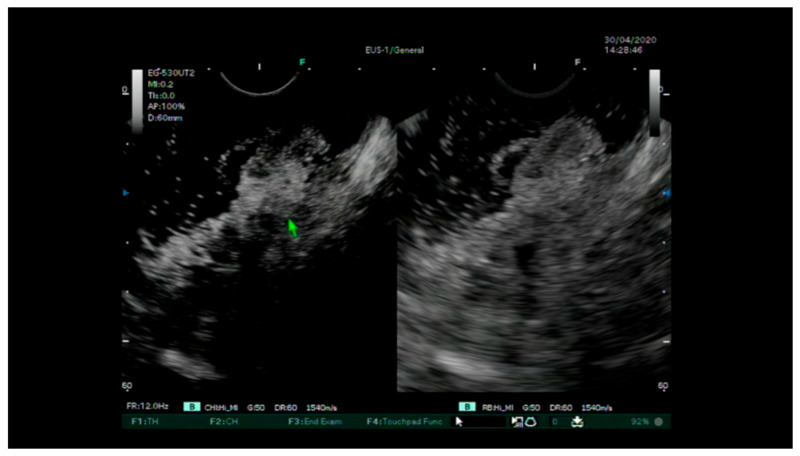
Evaluation of CH-EUS showing a focal point with interruption of the muscular layer corresponding to a focal adenocarcinoma at the final staging.

**Figure 3 diagnostics-13-02267-f003:**
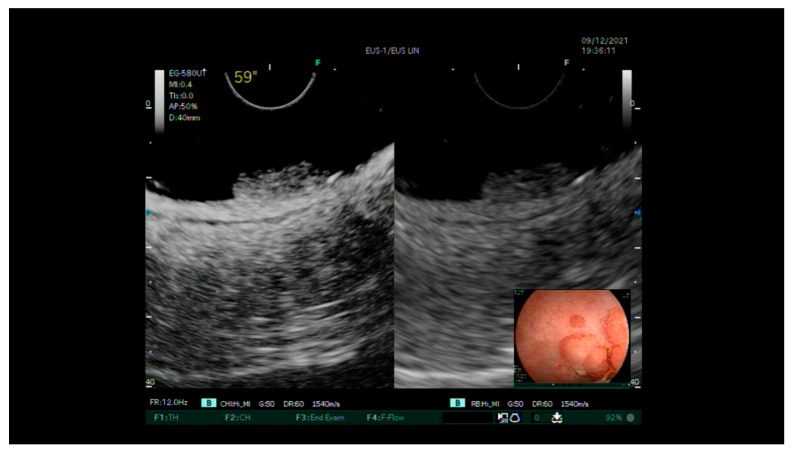
Evaluation of an LST-G type with dominant nodule showing integrity of the layers beyond the submucosa.

**Figure 4 diagnostics-13-02267-f004:**
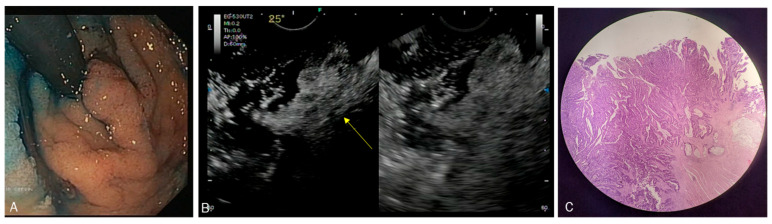
(**A**) Circumferential rectal lesion with a nodule suspected of invasive cancer. (**B**) CH-EUS showing a single point of invasive pattern. (**C**) Low-grade adenocarcinoma arising in tubulo-villous adenoma with high-grade dysplasia. Focal invasion of muscularis propria, pT2.

**Figure 5 diagnostics-13-02267-f005:**
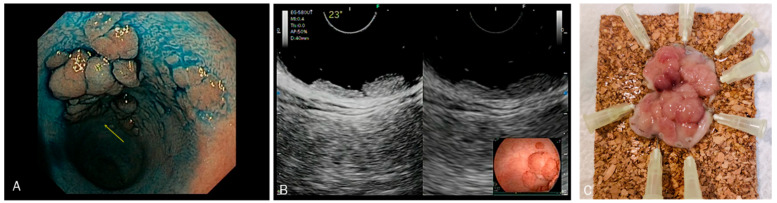
(**A**) 35 mm LST-G with a central nodule suspected for JNET type 3 (**B**) CH-EUS showing conservation of submucosal layers. (**C**) Lesion treated with Endoscopic Submucosal Dissection (ESD) and diagnosis of tubulovillous adenoma with high-grade dysplasia.

**Table 1 diagnostics-13-02267-t001:** Baseline characteristics of included population.

N	Sex	Age	Location ^1^	Size, mm	Paris and/or LST Classification	JNET Classification	Biopsy	CH-EUS Stage	MRI Stage	Treatment	pTNM Stage
1	M	74	Proximal	30	0-Is	Type 3 in the central area	Tubular adenoma HGD	uT2N+	T2/T3	RAR	pT2N0G1LV0R0
2	F *	68	Medium-distal	180	0-IIa + Is/LST-G nodular mixed	2b/focal 3 in dominant nodule	Tubular adenoma HGD	uT1aN0	T1/T2	ESD converted to RAR	pT2N1b
3	M	79	Medium-distal	60	0-IIa + Is/LST-G nodular mixed	2b/focal 3	-	uT1N0	-	Hybrid-ESD	pT1a G1
4	M	74	Proximal	20	0-Is	2b/focal 3	Adenoma HGD	uT2N0	T2/T3	RAR	pT2 N0 G2 LV0R0
5	F	70	Distal	30	0-Is + IIa/LST-G nodular mixed type	2b/focal 3 in dominant nodule	Villous adenoma HGD	uT1N0	-	ESD	Villous Adenoma HGD
6	M	65	Distal	35	0-Is + IIa/LST-G nodular mixed type	2b/focal 3 in dominant nodule	-	uTisN0	-	ESD	TV Adenoma HGD
7	F	65	Medium-distal	45	0-Is + IIa/LST-G nodular mixed type	2b/focal 3	-	uTis	-	Hybrid-ESD	TV Adenoma HGD
8	F	72	Distal	35	0-Is + IIa/LST-G nodular mixed type	2b/focal 3	Previous piecemeal resection: TV adenoma HGD	uT1aN0	-	ESD	TV Adenoma HGD
9	F	68	Medium-distal	80	0-IIa + Is/LST-G nodular mixed type	2b/focal 3 in 20 mm-dominant nodule	-	uTis	-	Hybrid-ESD	TV Adenoma HGD
10	M	81	Proximal	35	0-Is	2b/focal 3	-	uTis	-	Hybrid-ESD	TV Adenoma HGD
11	F	72	Medium-distal	30	0-IIa + IIc/LST-NG	2b/focal 3	Adenoma HGD and focal adenocarcinoma	uT1N0	T2	RAR + TME	pT1 N1b G2 LV0 R0
12	F	57	Distal	30	0-IIa + IIc/LST-NG	2b/focal 3	-	uT1N0	-	ESD	pT1a G1

^1^ Location considering rectum. RAR: rectal anterior resection; TME: trans-mesorectal excision; ESD: endoscopic submucosal dissection; TV: tubulo-villous; LGD: low-grade dysplasia; HGD: high-grade dysplasia. * Mckittrick–Wheelok Syndrome [28].

## Data Availability

Data are available if required.

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
