# Peer review of "Usefulness of Contrast-Enhanced Endoscopic Ultrasound (CH-EUS) to Guide the Treatment Choice in Superficial Rectal Lesions: A Case Series"

_diagnostics, 2023, doi:10.3390/diagnostics13132267_

Round 1

Reviewer 1 Report

Dear Dr. Gibiino & the team,

I had the pleasure in reading the manuscript, "Usefulness of Contrast-Enhanced Endoscopic Ultrasound (CH-2 EUS) to guide the treatment choice in superficial rectal lesions: 3 A Case Series". The study comes from Italy on a series of 12 subjects. 

The study is a retrospective analysis between Jan 2020 to Jan 2023.  The contrast EUS was performed by intravenous Sonovue contrast agent. The study is interesting and very helpful to the readers. I have suggested the authors to add a couple of images of the contrast-enhanced ultrasound.

The study could be published.

Kind regards,

I had the pleasure in reading your manuscript "Usefulness of Contrast-Enhanced Endoscopic Ultrasound (CH-2 EUS) to guide the treatment choice in superficial rectal lesions: 3 A Case Series. May I suggest that you add a couple of contrast-enhanced ultrasound images and this would help the readers to understand better. 

Best wishes.

Good

Author Response

We thank the reviewer for his comments. We added two figures showing the CH-EUS pattern at the evaluation of focal infiltration of adenocarcinoma (Figure 2) or alternatively a lesion not showing deep invasion (Figure 3).

Reviewer 2 Report

The manuscript entitled as "Usefulness Of Contrast-Enhanced Endoscopic Ultrasound (CH-EUS) to guide the treatment choice in superficial rectal lesions: A Case Series" is an interesting and well-structured Case Series that will be of interest to the readers of this journal. Also, the studies undertaken supports the results as well. But there are few things to be addressed in the manuscript. After thorough review, following are my observations:

1.       The abstract should be revised and summarized in the paper the way as it should be and conclusion should be added in it. The abstract should be uniform.

2. Methodology is quite obscure, should be restructured for proper understanding with technical approach   

3. Conclusion part needed to be more elaborated on outcome rather than findings.

4.Future perspectives are needed to elucidate in conclusion for the need this study.

Minor editing of English language required

Author Response

We thank the reviewer for his comments, and we provide point-by-point response:

  1. We modified the abstract divided into introduction, material and methods, results and conclusion. Now we think it summarises the manuscript correctly.
  2. We added details about the methodology. We hope that now the technical approach can be understood.
  3. We modified the conclusion paragraph, adding details about the outcome.
  4. We also added some considerations about future perspectives.

We revised the manuscript with respect to the English language and added some minor corrections.

Reviewer 3 Report

The concept of the article is interesting, and the authors used a pivotal and novel approach as the treatment choice in superficial rectal lesions.

I believe that the manuscript should be considered for publication.

Minor editing of English language required

Author Response

We thank the Reviewer for his comments and we improved the quality of English Language by minor revisions.

Round 2

Reviewer 2 Report

The authors have addressed all my queries. I recommend the manuscript for its possible publication in its present form.

Minor editing of English language required